# Prolonged Thermal Relaxation of the Thermosetting Polymers

**DOI:** 10.3390/polym13234104

**Published:** 2021-11-25

**Authors:** Alexander Korolev, Maxim Mishnev, Nikolai Ivanovich Vatin, Anastasia Ignatova

**Affiliations:** 1Department of Building Construction and Structures, South Ural State University, 454080 Chelyabinsk, Russia; 2Self-Healing Structural Materials Laboratory, Peter the Great St. Petersburg Polytechnic University, 195251 Saint Petersburg, Russia; 3Department of Technical Mechanics, South Ural State University, 454080 Chelyabinsk, Russia

**Keywords:** polymer, elasticity, deformability, modulus of elasticity, thermo-relaxation, glassing onset, thermal load

## Abstract

The rigidity of structures made of polymer composite materials, operated at elevated temperatures, is mainly determined by the residual rigidity of the polymer binder (which is very sensitive to elevated temperatures); therefore, the study of ways to increase the rigidity of polymer materials under heating (including prolonged heating) is relevant. In the previous research, cured thermosetting polymer structure’s non-stability, especially under heating, is determined by its supra-molecular structure domain’s conglomerate character and the high entropy of such structures. The polymer elasticity modeling proved the significance of the entropy factor and layer (EPL) model application. The prolonged heating makes it possible to release adsorptive inter-layer bonds and volatile groups. As a result, the polymer structure is changing, and inner stress relaxation occurs due to this thermo-process, called thermo-relaxation. The present study suggests researching thermo-relaxation’s influence on polymers’ deformability under load and heating. The research results prove the significant polymer structure modification due to thermo-relaxation, with the polymer entropy parameter decreasing, the glassing onset temperature point (Tg) increasing by 1.3–1.7 times, and the modulus of elasticity under heating increasing by 1.5–2 times.

## 1. Introduction

According to most of the research, elasticity of thermosetting polymers under decreasing heat is a result of the following processes:Polymer molecules thermo-expanding with their morphology and pack changes [1,2,3,4,5].Inter-molecular bonds’ flexibility and torsion increasing under heating [6,7,8].

The previous research [9] presented structural and mass changes under heating. The thermo-gravimetry proved the continuous mass loss of absolutely cured epoxy and phenolic polymers under all heating temperatures. Therefore, the processes under heating are complex and have not been researched enough.

The application of the layer model in elasticity prediction ensured the proposal of a domain-like polymer structure, which appeared due to polymerization promotion from the most dispersed reactional centers and displacement of the most volatile and unreactive fractions on the peripheral surfaces of hardening domains. The domains dominate inter-molecular or van-der-Waals bonds. Domains form the border between phases of the interphase transition zone (ITZ). The ITZ is a consequence of the conglomerate type of polymer structure and is realized because of supramolecular processes during polymer glassing. Previous studies have revealed the following ITZ conditions:Polymer glassing is an insular distributed process developing from reaction centers. Structures found due to this research process are globular or domain conglomerate structures, because the peripheric surface of domains has properties other than inner [10,11].Inter-domain bonds are molecular, so they have less energy and are more moveable under load and heating [12,13,14,15]. The energy of dissipation or entropy can determine the energy of these bonds.Molecular inter-domain bonds determine the rheology of ITZ and make its character viscose or viscoelastic [16,17,18,19,20,21]. It reflects on the viscoelastic deformative properties of the overall polymer structure. ITZ is an inter-domain surface and matrix of the glassed conglomerate. Due to its matrix character, the ITZ is endless and can mass remove and create sliding surfaces for deformation under load and heating.

With temperature increases, inter-domain bonds of polymer structure lose elasticity, and polymer deformations become viscoelastic [21,22,23,24]. A curve illustrating the viscoelastic behavior of thermosetting polymers is presented, for example, in [21]. The parameter characterizing the transition from elastic to viscoelastic condition is the glassing onset temperature point, Tg. Along with heating, the mass loss of weakly bonded components occurs simultaneously, so the final structure and its parameters are changing. The stress–strain condition of the structure is changing too, which influences the thermo-relaxation process. Many researchers are dedicated to studying the thermo-relaxation of polymers [25,26,27,28,29,30,31,32,33,34] using Arrhenius, Vogel-Fulcher-Tammann (VFT), and Bässler laws, as thermodynamic equations based on free energy parameters, but most describe cycle processes with viscosity, rest deformation, and strength definition in the glassing temperature Tg area. In construction practice, the actuality has the polymer deformability and the influence of thermo-relaxation on it in all areas, from normal to Tg temperatures.

Therefore, this research explores polymers’ deformability under heating prediction, where the Tg is an important parameter for exploiting the temperature limit. The presented research aims to determine the influence of thermo-relaxation on the polymer glassing onset temperature point and the modulus of elasticity under heating. The research object was a glassed polymer with epoxy, phenolic, and epoxy-phenolic glassed binders and similar materials.

The subject of this research is the glassed polymer modulus of elasticity under heating and its glassing onset temperature after the previous step-by-step definite temperatures of heating of polymer samples.

The research includes several objectives, as follows:Thermo-gravimetry and modulus of elasticity after heating of glassed polymers.DMA research for glassing onset temperature points before and after thermo-relaxation.Study of the polymer “deformation modulus-temperature” dependence before and after thermo-relaxation.

## 2. Materials and Methods

### 2.1. Materials

Table 1 presents the characteristics of thermosetting polymer materials. The thermal stability (as weight loss) and deformability were measured at elevated temperatures for these materials. The table shows the percentage by mass of components in the composition of the binder.

The components described below were used to make the binders:Epoxy resin KER 828, with the following main characteristics: Epoxy Group Content (EGC) 5308 mmol/kg, Epoxide Equivalent Weight (EEW) 188.5 g/eq, viscosity at 25 °C 12.7 Pa.s, HCl 116 mg/kg, and total chlorine 1011 mg/kg. Manufacturer: KUMHO P&B Chemicals, Gwangju, South Korea.Hardener for epoxy resin methyl tetrahydrophthalic anhydride with the following main characteristics: viscosity at 25 °C 63 Pa.s, anhydride content 42.4%, volatile fraction content 0.55%, and free acid 0.1%. Manufacturer: ASAMBLY Chemicals company Ltd., Nanjing, China.Alkofen (epoxy resin curing accelerator) with the following main characteristics: viscosity at 25 °C 150 Pa.s, molecular formula C_15_H_27_N_3_O, molecular weight 265, and amine value 600 mg KOH/g. Manufacturer: Epital JSC, Moscow, Russian Federation.Resol phenolic resin SFRZ-309 with the following main characteristics: viscosity at 25 °C 700 mPa.s, not more than 20% (*m*/*m*) water, and not more than 20% (*m*/*m*) free phenol. Manufacturer: FCP “Sverdlov Plant”, Dzerzhinsk, Russian Federation.

The components were mixed in the above proportions at room temperature of about 25 °C. Mixing to a homogeneous consistency was carried out mechanically with an electric drill with a mixing attachment.

Epoxy resin samples were cured at 120 °C for 30 min, epoxy-phenolic binder was cured at 70 °C for 120 min, then 120 °C for 60 min, and phenolic resin was cured at 70 °C for 90 min, then at 120 °C for 60 min. After curing, all samples were kept at 150 °C for 12 h.

The samples were cured in silicone molds in the form of plates, from which bar samples were subsequently cut. A general view of a silicone mold with curing binders for making samples and cured samples of binders is shown in Figure 1.

### 2.2. Methods

After curing, bar samples of cured polymer binders were exposed to prolonged exposure at elevated temperatures. In this case, one of the samples of each type was a control one, which was not exposed to aging at elevated temperatures. Before the start of long-term storage, the initial glass transition temperature was determined for all samples using DMA, and the elastic modulus was determined using three-point bending at room temperature (22–25 °C), which will hereinafter be referred to as the “cold” modulus of elasticity, and at elevated temperatures.

Long-term exposure of samples at elevated temperatures was carried out according to the following program: 189 h at 160 °C, 182 h at 190 °C, and 429 h at 220 °C. Periodically (after 6–48 h), the samples were cooled at a rate of about 1 °C per minute up to 50 °C, were taken out of the laboratory oven, weighed, and after each weighing (TG analysis), their elastic modulus was determined at room temperature (22–25 °C).

After the end of long-term holding at elevated temperatures, the glass transition temperatures were determined for each sample using DMA, and the curves of the dependence of the elastic modulus in bending on temperature were plotted. The dynamic mechanical analysis (DMA) was performed using a NETZCSH DMA 242C analyzer (Erich NETZSCH GmbH & Co. Holding KG, Gebrüder-Netzsch-Straße 19, Germany).

During the tests, the dynamic modulus of elasticity, the modulus of viscosity, and the tangent of the angle of mechanical losses were determined. When the polymer reaches the glass transition temperature (Tg), a significant drop in the dynamic modulus of elasticity is observed. Following GOST R 57739-2017 [35], the glass transition temperature was determined as the intersection of two tangents to the dynamic modulus of the elasticity graph.

Table 2 shows the sizes of the samples tested. The samples were subjected to sinusoidal force action according to the three-point bending scheme at a span of 20 mm to determine the viscoelastic characteristics and glass transition temperature. The frequency was 5 Hz, the value of the static component of the load was 0.72 N, and the dynamic component of the load was 0.4 N. The experiments were carried out in the temperature range of 25–130/140 °C at a rate of 1 °C/min. For sample No. 3, tests were carried out in the temperature range from 25 to 300 °C.

As a result, the values of the glass transition temperatures were determined for the samples of polymer binders before and after prolonged exposure at elevated temperatures.

Along with modern high-precision methods of mechanical testing of polymer materials and composites (for example, those described in [36]), three-point bending tests (for example, described in [37]), which have long been known but have not lost their relevance due to the simplicity of the equipment used, are widely used. The elastic modulus at three-point bending was the main mechanical characteristic for estimating the stiffness of polymers at normal and elevated temperatures.

Three-point bending tests of polymer samples were carried out on a Tinius Olsen h100ku testing machine (Tinius Olsen GmbH, Goethestr. 7b, 86161 Augsburg, Germany) in a specially made small-sized chamber, which provides heating and maintains the temperature up to 300 °C.

According to the producer’s data for a Tinius Olsen h100ku machine, the load accuracy was ±0.5% in the range of 0.2–100% of the installed force sensor (100 kN). The resolution of measuring the crosshead movement was 0.1 mm, with an error of up to 0.01 mm. The sample center point displacement under the load was monitored by a mechanical dial gauge mounted on the bottom of the small-sized test chamber. This monitoring was aimed at excluding the machine compliance influence. The difference between the displacement readings along the traverse and the dial gauge did not exceed 2%.

Three-point bending tests determined the cured sample deformation modulus at temperatures from 25 to 230 °C. The tests were carried out following Russian State Standard GOST 25.604-82 [36]. The experimental values of elasticity modulus at the bending of the samples were determined at a 2 mm/min loading rate. The determination of the elasticity modulus was carried out under loading with two load steps.

When determining the elastic modulus in bending, the samples were preliminarily loaded with a concentrated force to the level of normal stresses of 1.2 MPa. Further, loading was carried out, and the determination of the elastic modulus was carried out at the range of normal stresses of 1.2–3.2 MPa.

The samples were preliminarily held at elevated temperatures until they were completely warmed up to the test temperature. The temperature during the tests was maintained by a thermostat and controlled by two thermocouples. One thermocouple measured the temperature on the surface of the bent specimen. The second thermocouple measured the temperature inside the control specimen, located next to the test specimen. The installation diagram is shown in Figure 2.

## 3. Results and Discussion

### 3.1. Polymer Mass Lost under Heating and Modulus of Elasticity after Heating Research

Empirical dependences of epoxy, epoxy-phenolic and phenolic polymers’ mass lost upon exposure at elevated temperatures are shown in Figure 3. Indexes 1 and 2 are the experimental repetition with the same compounds. It is shown that at each stage of heating, significant mass loss appears. The total mass lost can achieve 4–13%, and the most mass lost begins developing at 160 °C. The prolonged heating was performed in three stages, at 160, 190, and 220 °C. The mass loss continued at each stage until constant mass. With the transition to the next increasing temperature stage, the mass loss begins again at the next constant mass. This effect shows that the polymer structure keeps many weakly bonded volatile groups with different bond strengths. Since most of the mass loss is developing up to 220 °C, it is proposed that these groups have to dominate adsorptive or van-der-Waals bonds. Therefore, those volatile groups sublimate at each stage if heat energy is enough to destroy their adsorptive bonds. The duration of mass loss is related to the difficulty of mass removal from the inner to the outer space of the polymer volume. In this case, the thermo-relaxation process is realized, and polymer structure entropy decreases.

From this point of view, several volatile polymer groups between elastic domains define the viscoelasticity of the polymer under load and heating, particularly the glassing onset temperature point. It is proposed that with volatile groups decreasing after heating, the width of ITZ decreases too, and the strength of inter-domain bonds, elasticity, and Tg has to increase significantly. It is suggested by empirical dependences of epoxy, epoxy-phenolic, and phenolic polymers’ elasticity at normal temperature modulus after prolonged exposure to high temperatures (Figure 4).

In one of the samples of pure phenolic resin (PF 2-2), at the initial stage of thermal relaxation, the weight loss was more intense. The weight loss at this stage was probably mainly due to the evaporation of water that forms during curing. However, in the further stages, the intensity of weight loss leveled off for both samples. If we do not consider the initial section, then in the further stages in both samples, the process of weight loss during prolonged storage had a comparable intensity.

Additionally, the phenolic binder samples at a certain moment of exposure at a temperature of 220 °C simultaneously began to lose weight more intensively than before, while the temperature did not change (see Figure 3, PF 2-1 and PF 2-2 after 720 h of exposure). It is also noteworthy that after a change in the intensity of mass loss, the elastic modulus of both exposed samples (see Figure 5 after 720 h of exposure) also decreased, while the modulus of elasticity of the reference sample did not change. This suggests that in the structure of phenolic resin as a result of prolonged exposure to temperature, there was an accumulation of certain changes, which manifested themselves in the form of an increase in the intensity of weight loss and a decrease in hardness, without increasing the external temperature.

Empirical data prove that the normal temperature elasticity modulus of polymers after thermo-relaxation changes wavily, depending on temperature and time of exposure. It demonstrates that exposure under heating modifies the polymer structure, and the thermo-relaxation process is realized. Ultimately, however, the difference between the normal temperature modulus of elasticity before and after exposure does not exceed 10%, so it cannot be said that the normal temperature modulus of elasticity changed significantly after exposure.

Another situation arises when considering the glass transition temperature and elastic modulus at elevated temperatures (high-temperature elastic modulus).

### 3.2. Polymers before and after Thermo-Relaxation DMA

The results of the DMA are presented in Figure 6 and Figure 7.

As it can be seen from the DMA curves, a significant change as a result of thermo-relaxation is developing. Not only does Tg increase by 1.3–1.7 times, but the straight/rest dynamic modulus ratio (tan d) at the low peak decreased by 1.5 times, so the deformation modulus increases after Tg. It proves that the thermo-relaxation influences the polymer elasticity proposal.

### 3.3. Testing the Polymer Elasticity Modulus concerning Temperature and Entropy Factor after Thermo-Relaxation

The results of the elasticity modulus under heating tested before and after exposure at elevated temperatures are presented in Figure 8, Figure 9 and Figure 10.

The thermo-relaxation increases the polymers’ elasticity under heating. According to the DMA results, the breakpoint to viscoelasticity increases by up to more than 1.3 times.

According to the layer model (EPL model) [9], the previously proposed and tested modulus of deformation dependence on temperature can be calculated as:$$E_{T}=\frac{k_{pl}E_{0}}{{1-n}_{T}(1-k_{pl})}$$
where n_T_ is a fraction of elastic bonds in all bonds’ amount, and *k_pl_* is the plastic bonds’ deformability modulus to elastic bonds’ elasticity modulus ratio (deformability ratio).

Based on previous temperature-changing polymer structure research results, the deformability ratio of the adsorptive bond is related to temperature by the entropy equation:(2)$$k_{pl}=1-S*ln\frac{T_{s}}{T_{0}}$$
where *S* (J/J) is the coefficient of the bond entropy equal to the relation between entropy and potential energy of the elastic bond at standard temperature, and *S* and n_T_ are correlative parameters with physical essence in this research stage.

Before the thermo-relaxation, the entropy and structure coefficients are as follows:

For epoxy resin composition, the equation is:(3)$$k_{pl-epox}=1-3.6ln\frac{{273+t}_{s}}{298}$$
(4)$$E_{T}=\frac{k_{pl-epox}E_{0}}{1-0.77(1-k_{pl-epox})}$$

For phenolic resin composition:(5)$$k_{pl-fenol}=1-2.5ln\frac{{273+t}_{s}}{298}$$
(6)$$E_{T}=\frac{k_{pl-fenol}E_{0}}{1-0.65(1-k_{pl-fenol})}$$

For epoxy-phenolic resin composition:(7)$$k_{pl-ep-fen}=1-4.7ln\frac{{273+t}_{s}}{298}$$
(8)$$E_{T}=\frac{k_{pl-ep-fen}E_{0}}{1-0.8(1-k_{pl-ep-fen})}$$

After the thermo-relaxation, the entropy and structure coefficients are as follows:

For epoxy resin composition, the equation is:(9)$$k_{pl-epox}=1-2.8ln\frac{{273+t}_{s}}{298}$$
(10)$$E_{T}=\frac{k_{pl-epox}E_{0}}{1-0.77(1-k_{pl-epox})}$$

For phenolic resin composition:(11)$$k_{pl-fenol}=1-1.6ln\frac{{273+t}_{s}}{298}$$
(12)$$E_{T}=\frac{k_{pl-fenol}E_{0}}{1-0.65(1-k_{pl-fenol})}$$

For epoxy-phenolic resin composition:(13)$$k_{pl-ep-fen}=1-2.3ln\frac{{273+t}_{s}}{298}$$
(14)$$E_{T}=\frac{k_{pl-ep-fen}E_{0}}{1-0.8(1-k_{pl-ep-fen})}$$

Table 3, Table 4 and Table 5 present the data of the actual and calculated deformation modulus under heating before and after thermo-relaxation.

The average derivation of the elasticity modulus under heating calculation is just 2.6%. It proves that layer modeling included the highest accuracy and applicability entropy factor. Models’ analysis after thermo-relaxation showed that the entropy coefficient decreased by 1.5–2 times, but the structure layer coefficient remained constant. This means that the thermo-relaxation result is the reduction of the structure entropy and elasticity under heating, and the increase of the glass point.

## 4. Conclusions

As a result of this analytical and experimental study, a significant modification of polymer structure after prolonged heating was determined. The modification was determined by a significant change of the Tg and deformation DMA modulus. After a successful layer model of polymers under load and prolonged heating elasticity prediction testing, the entropy before and after prolonged heating was determined. The results proved that entropy of the polymer structure after prolonged heating was significantly reduced, by 1.5–2 times. It points out that under prolonged heating under temperatures of more than the Tg, the polymer thermo-relaxation was realized.

The high effectiveness of the thermo-relaxation method for polymers under an increased heating elasticity modulus was proven. Thermo-relaxation, realized as long-term secondary heating of polymer samples, significantly improved polymers’ properties and structure, such as:Volatile fractions’ mass loss in the range of 5–10%.The glassing onset point increased by 1.3–1.7 times.The normal temperature modulus of elasticity in bending after prolonged exposure at elevated temperatures did not change significantly. In contrast, the moduli of elasticity at high temperatures for all types of binders increased by several times.

This method can be used in practice if the first heating of structures by the recommended curves is realized.

The original layer-bonded model (EPL model) with thermodynamic parameters has been proposed and proven in the calculation of polymers’ deformation modulus depending on the temperature under heating in a normal to Tg temperature range for all kinds of thermosetting polymers. Application of this model can be useful in calculating stresses in heated polymer-reinforced structures and developing methods of increasing heat resistance for polymer composites.

## Figures and Tables

**Figure 1 polymers-13-04104-f001:**
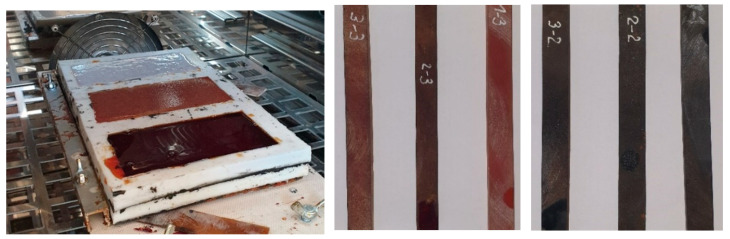
Samples in the curing process and after cutting from cured plates.

**Figure 2 polymers-13-04104-f002:**
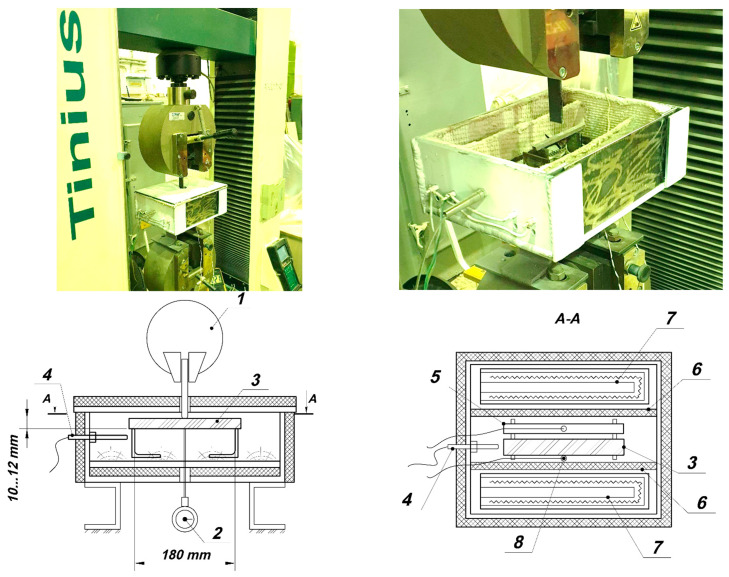
Heating three-point bending test rig: 1—Testing machine, 2—Indicator, 3—Bend specimen, 4—Thermocouple of a thermostat, 5—Thermocouple specimen, 6—Baffle, 7—Heater, 8—Thermocouple.

**Figure 3 polymers-13-04104-f003:**
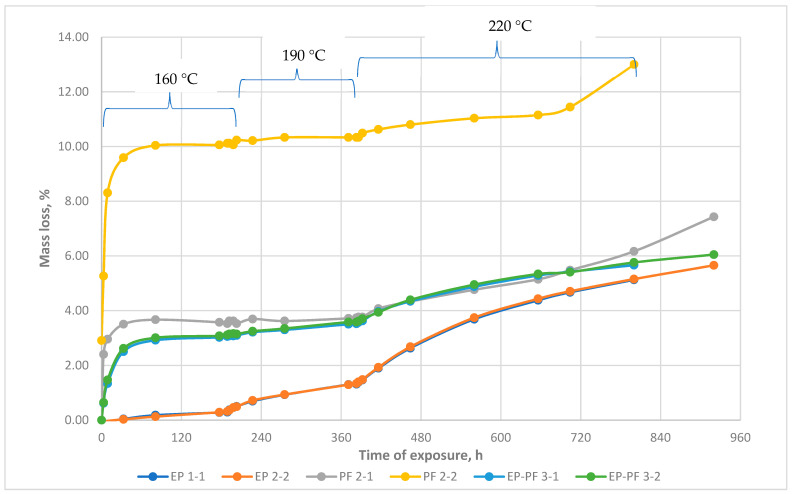
Character dependence of polymer mass loss of polymer samples on temperature: epoxy resin (EP), phenolic resin (PF), and epoxy-phenolic (EP-PF).

**Figure 4 polymers-13-04104-f004:**
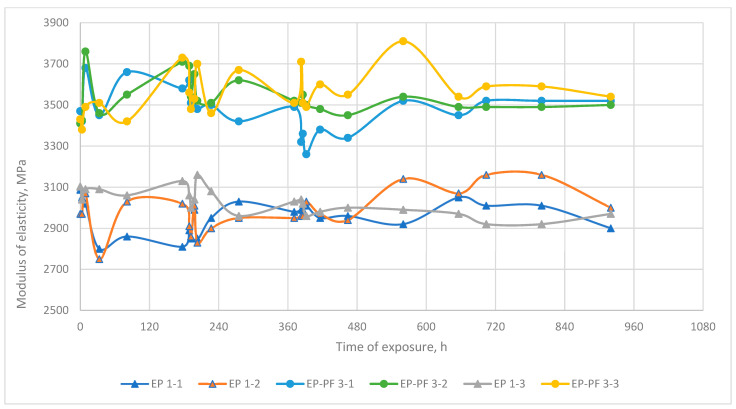
Character dependence of polymer elasticity (cold) modulus after exposure at elevated temperatures: EP—epoxy resin, EP-PF—epoxy-phenolic. EP 1-1, 1-2, EP-PF 3-1, 3-2: exposure at elevated temperature, EP 1-3, EP-PF 3-3: not exposed at elevated temperature.

**Figure 5 polymers-13-04104-f005:**
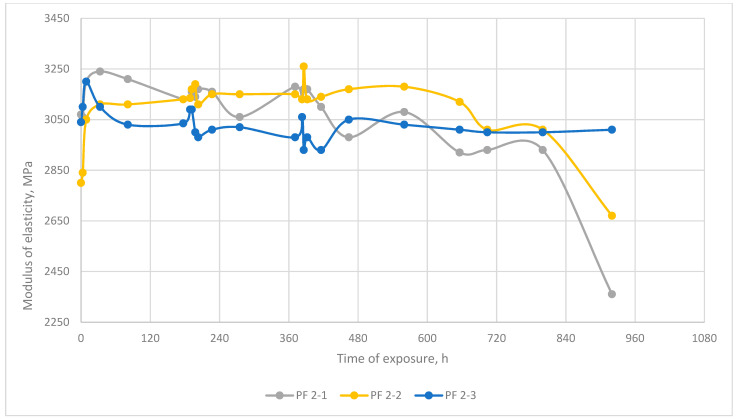
Character dependence of phenolic resin elasticity (cold) modulus after exposure at elevated temperatures. PF 2-1, PF 2-2: exposure at elevated temperature, PF 2-3: not exposed at elevated temperature.

**Figure 6 polymers-13-04104-f006:**
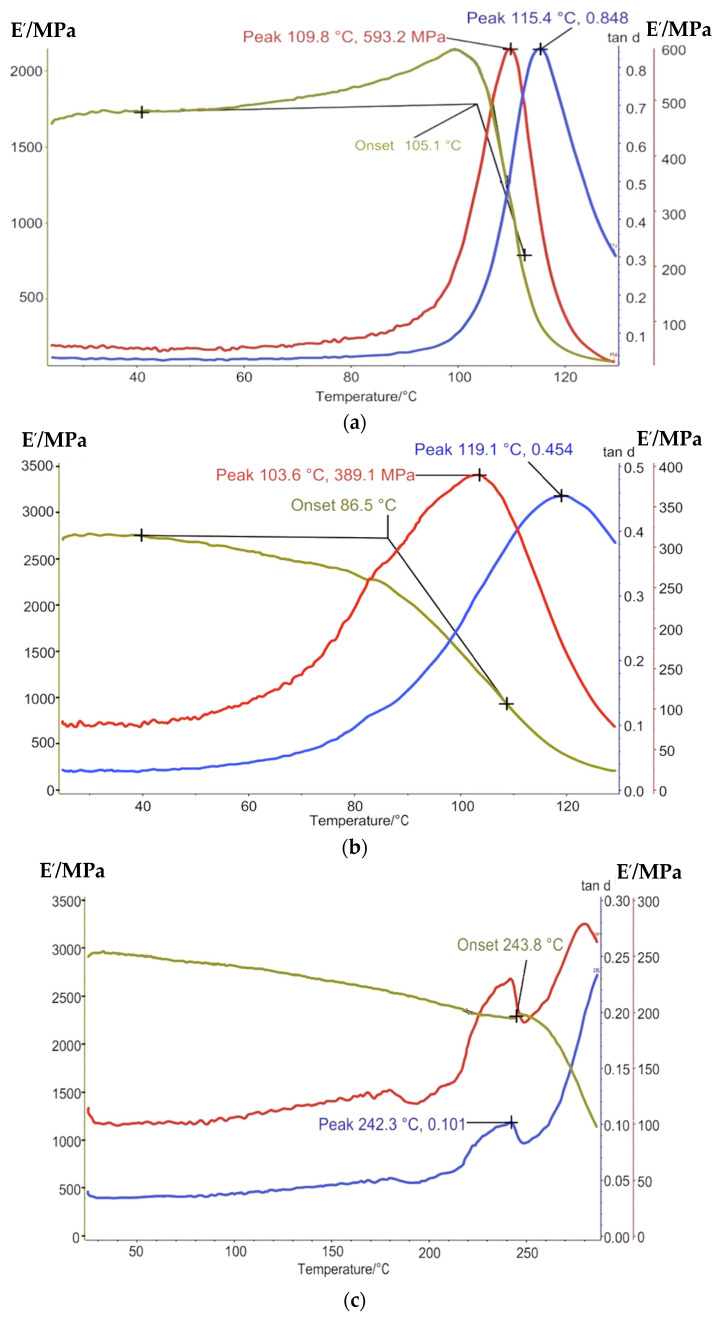
DMA curves of glassed polymers before thermo-relaxation: (**a**) epoxy resin, (**b**) epoxy-phenolic resin, and (**c**) phenolic resin.

**Figure 7 polymers-13-04104-f007:**
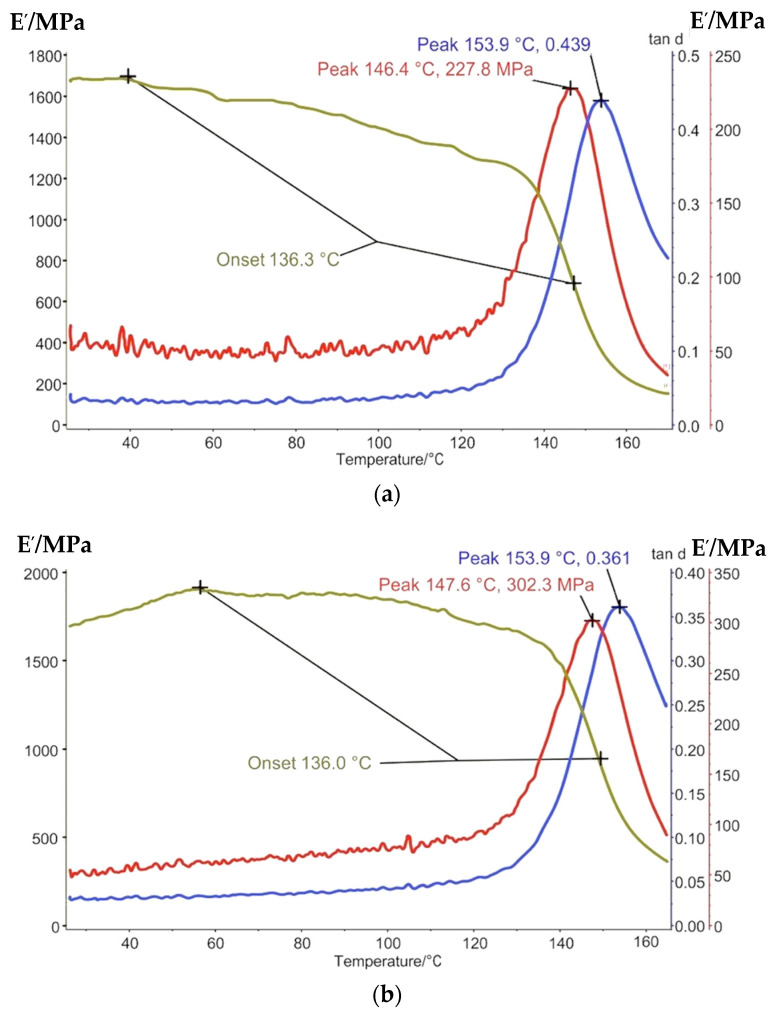
DMA curves of glassed polymers before thermo-relaxation: (**a**) epoxy resin and (**b**) epoxy-phenolic resin.

**Figure 8 polymers-13-04104-f008:**
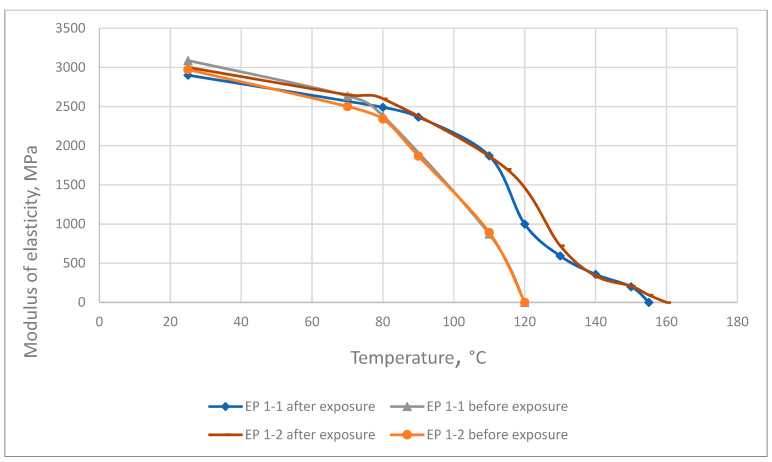
Dependence of the epoxy resin elastic modulus on temperature before and after long exposure at elevated temperatures.

**Figure 9 polymers-13-04104-f009:**
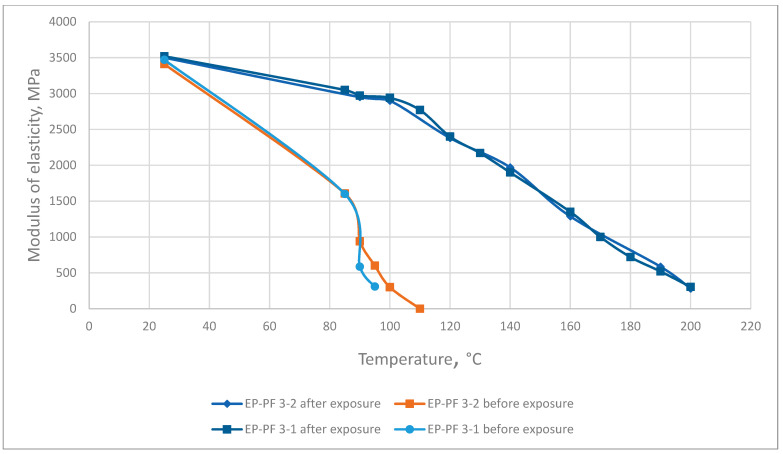
Dependence of the epoxy-phenolic resin elastic modulus on temperature before and after long exposure at elevated temperatures.

**Figure 10 polymers-13-04104-f010:**
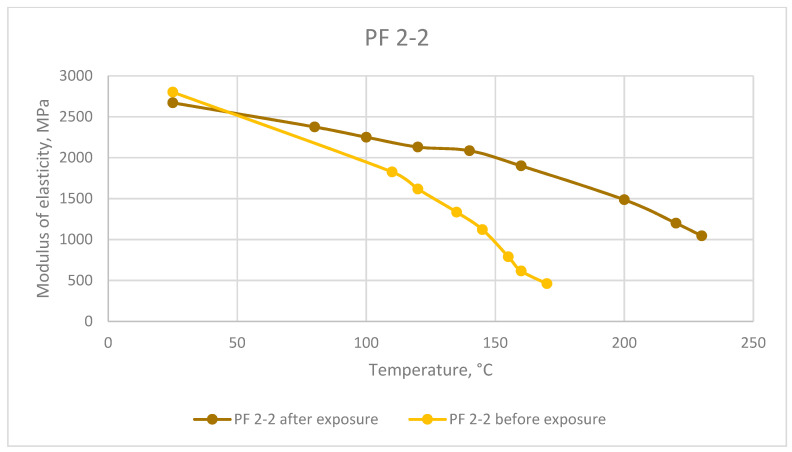
Dependence of the phenolic resin elastic modulus on temperature before and after long exposure at elevated temperatures.

**Table 1 polymers-13-04104-t001:** Types of binders investigated.

No.	Compound	Designation	Thermogravimetric Analysis	Three-Point Bending Test at Elevated Temperature
1	Epoxy resin (Ker 828 52.5% + MTHPA 44.5% + alkofen 3%)	EP	+	+
2	Phenolic resin (SFZ-309) 100%	PF	+	+
3	Epoxy-phenolic resin (KER 828 45% + SFZ-309 55%)	EP-PF	+	+

**Table 2 polymers-13-04104-t002:** Sample sizes for DMA.

No.	Compound	Width, mm	Thickness, mm
1	Epoxy resin (Ker 828 52.5% + MTHPA 44.5% + alkofen 3%)	8.72	3.48
2	Phenolic resin (SFZ-309) 100%	5.95	2.8
3	Epoxy-phenolic resin (KER 828 45% + SFZ-309 55%)	7.03	2.95

**Table 3 polymers-13-04104-t003:** Comparative actual and calculated data for epoxy modulus of deformation under heating before and after thermo-relaxation.

Compound	Temperature, °C	E_fact_, MPa	S, J/J	*k_pl_*	E_calc_, MPa	% Derivation
Epoxy before thermo- relaxation	25	3028	3.6	1.000	-	-
	70	2567		0.494	2450	−4.6
	80	2360		0.390	2227	−5.6
	90	1868		0.290	1899	1.7
	110	893		0.097	960	7.5
Epoxy after thermo- relaxation	25	2950	2.8	1.000	-	
	70	2650		0.606	2610	−1.5
	80	2600		0.523	2443	−6.0
	90	2365		0.448	2259	−4.4
	110	1870		0.297	1878	0.4
	115	1700		0.261	1817	6.9
	120	1000		0.225	1618	61.8
	130	724		0.154	1307	80
	140	338		0.086	858	153
	150	208		0.019	231	6.3

**Table 4 polymers-13-04104-t004:** Comparative actual and calculated data for phenolic modulus of deformation under heating before and after thermo-relaxation.

Compound	Temperature, °C	E_fact_, MPa	S, J/J	*k_pl_*	E_calc_, MPa	% Derivation
Phenolic before thermo-relaxation	25	2935	2.5	1	-	-
	110	1823		0.372	1846	1.3
	120	1678		0.308	1644	2.0
	135	1354		0.214	1286	−5.0
	145	1030		0.154	1004	−2.5
	155	672		0.095	676	0.6
	160	471		0.065	491	4.2
	170	462		0.009	74	−84
Phenolic after thermo- relaxation	25	2515	1.6	1	-	-
	80	2375		0.729	2362	−0.5
	100	2250		0.641	2232	−1.2
	120	2130		0.557	2089	−1.9
	140	2085		0.478	1931	−7.3
	160	1900		0.402	1756	−7.5
	200	1486		0.261	1340	−9.8
	220	1200		0.210	1155	−3.8
	230	1045		0.178	1022	−2.2

**Table 5 polymers-13-04104-t005:** Comparative actual and calculated data for epoxy-phenolic modulus of deformation under heating before and after thermo-relaxation.

Compound	Temperature, °C	E_fact_, MPa	S, J/J	*k_pl_*	E_calc_, MPa	% Derivation
Epoxy-phenolic before thermo-relaxation	25	3440	4.7	1	-	-
	85	1603		0.138	1528	−4.7
	90	763		0.072	968	26.8
	95	455		0.008	139	−69
	100	300		-	-	-
Epoxy-phenolic after thermo-relaxation	25	3510	2.3	1	-	-
	85	3050		0.578	3071	0.7
	90	2960		0.546	3010	1.7
	100	2920		0.483	2887	−0.8
	110	2770		0.423	2765	−0.2
	120	2390		0.364	2420	1.2
	130	2170		0.305	2190	0.9
	140	1935		0.249	1580	−18.3
	160	1320		0.141	1147	−13.1

## Data Availability

The data presented in this study are available on request from the corresponding author.

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
