# Peer review of "Prolonged Thermal Relaxation of the Thermosetting Polymers"

_polymers, 2021, doi:10.3390/polym13234104_

Round 1

Reviewer 1 Report

This study conducted a series of experiments on the glassy polymer systems with epoxy and phenolic binders. The influence of thermo-relaxation on the initial glass transition temperature and modulus upon heating are systematically characterized. Meanwhile, the dependence of the deformation modulus on the temperature before and after thermo-relaxation are further studied using the layer-bonded model, which is claimed to provide an effective guidance on calculating stresses in polymer reinforced structures undergoing the viscoelastic transition. On the whole, this manuscript is well presented, and the relevant experimental results and modeling analysis are rational. However, some questions should be addressed. The detailed comments are listed as follows.

1. In the abstract section, the research background should be brief.

2. A diagram is needed to illustrate the viscoelastic relaxation mechanism of amorphous polymers.

3. In figures (7) and (8), subgraph of (a) and (b) are suggested in a horizontal arrangement.

4. This study claim of the Tg is determined by the heating rate. The VFT model and the effect of polar interaction on Tg should be covered in the Introduction, as in Refs. (1) Haibao Lu and Wei Min Huang. On the origin of the Vogel-Fulcher-Tammann law in the thermo-responsive shape memory effect of amorphous polymers. Smart Materials & Structures. 2013, 22(10): 105021. (2) Haibao Lu, Jinsong Leng and Shanyi Du. A Phenomenological Approach for the Chemo-Responsive Shape Memory Effect in Amorphous Polymers. Soft Matter. 2013, 9(14): 3851-3858.

5. Is the measurement method of the entropy of polymer structure reliable?

6. The conclusion section merely states the results of the study, and the scientific significance behind these results are not sublimated.

Author Response

Dear Reviewer, we attach a file with answers to your comments

Reviewer 2 Report

Manuscript ID: Polymers-1433008

“Prolonged Thermal Relaxation of the Thermosetting Polymers”

Here, the authors proposed an analytical and experimental study to increase the thermo-relaxation of polymer matrices under heating in this study. Three different resins were tested: epoxy, phenolic resin, and epoxy-phenolic resin. The findings confirmed an increase in the glassing onset temperature point due to a change in polymer structure.

The work is overall positive, even if the quality of some images should be improved, and more details on thermal characterization (TGA, DSC) should be provided. In the abstract, the description of previous and current work should be balanced; and some aspects of the introduction should be highlighted regarding the purpose of the analyzed materials, the reason for the chosen characterization technique, and potential applications.

I propose this manuscript to be considered for publication in Polymers after major revision. Other specific suggestions are as follows:

-lines 78-80        “The experimental research was carried out at the South Ural State University, Chel-79 yabinsk, the Russian Federation”, I don't think it's necessary to specify it here. The authors' affiliation is already stated at the start;

- who are the suppliers of the reagents?

-please explain the acronyms  EGC (line 87) EEW (line 88) HCl (line 88) KOH (line 95)

- Figure 2 is a little hazy. Images in black and white are barely visible. I would recommend that black and white images be replaced with color images. I would suggest using letters to define each element in Figure 2 and recalling the description of each element in the caption. The schematization of the apparatus in Figure 2 is hazy. Where was this schematization taken from?

- lines 126-130        Please, describe the thermal cycle to which the sample was subjected during the TGA analysis, specifying the temperature range and the heating rate.

-Does it appear that DSC analysis has been done? (Materials and Methods line 130). Where are the data shown?

- in Table 1, the designation of the formulations has been presented in table 1. Then, which formulations in Figure 3 correspond to EP 1-1, EP 2-2, PF 2-1, PF 2-2, EP-PF 3-1, EP-PF 3-2? Are they repetitions of tests on the same samples? Please, specify.

- All of the other tested samples' curves are superimposable, but there is a significant difference between PF 2-1 and PF 2-2. Could you please explain why?

- the data shown in Figure 6 should be reprocessed and represented with the same format as the others (Excel). In this format, they are difficult to read.

In general, please also use another way to express without using dots

-“increasing to 1,3…1,7 times”(line 21)

-“using three-point bending at room (22... 25 ° C) temperature” (line 115)

-“was determined at room (22... 25 ° C) temperature” (line 122)

-“size about 1…2 mm in cross-section” (line 128)

-“range of normal stresses 1.2...3.2 MPa” line 178

-“Not only GT in 1.3…1.7 times increase” line 236

etc…..please check all the text.

Author Response

(The authors gave the same response as above.)

Round 2

Reviewer 2 Report

Overall, the work is positive, and the suggestions have been taken into consideration. I propose the manuscript be published in its current form.

Author Response

Dear reviewer, thank you for your review